# Dissecting Adaptation Mechanisms to Contrasting Solar Irradiance in the Mediterranean Shrub *Cistus incanus*

**DOI:** 10.3390/ijms20143599

**Published:** 2019-07-23

**Authors:** Federico Sebastiani, Sara Torre, Antonella Gori, Cecilia Brunetti, Mauro Centritto, Francesco Ferrini, Massimiliano Tattini

**Affiliations:** 1Institute for Sustainable Plant Protection (IPSP), The National Research Council of Italy (CNR), 50019 Sesto Fiorentino (Florence), Italy; 2Department of Agriculture, Food, Environment and Forestry, University of Florence, 50019 Sesto Fiorentino (Florence), Italy; 3Institute of BioEconomy, The National Research Council of Italy (CNR), 50019 Sesto Fiorentino (Florence), Italy

**Keywords:** excess solar irradiance, flavonoids, Mediterranean climate, photosynthesis, PSII functionality, transcriptomics, shade, violaxanthin cycle pigments, water relations

## Abstract

Molecular mechanisms that are the base of the strategies adopted by Mediterranean plants to cope with the challenges imposed by limited or excessive solar radiation during the summer season have received limited attention. In our study, conducted on *C. incanus* plants growing in the shade or in full sunlight, we performed measurements of relevant physiological traits, such as leaf water potential, gas exchange and PSII photochemistry, RNA-Seq with *de-novo* assembly, and the analysis of differentially expressed genes. We also identified and quantified photosynthetic pigments, abscisic acid, and flavonoids. Here, we show major mechanisms regulating light perception and signaling which, in turn, sustain the shade avoidance syndrome displayed by the ‘sun loving’ *C. incanus*. We offer clear evidence of the detrimental effects of excessive light on both the assembly and the stability of PSII, and the activation of a suite of both repair and effective antioxidant mechanisms in sun-adapted leaves. For instance, our study supports the view of major antioxidant functions of zeaxanthin in sunny plants concomitantly challenged by severe drought stress. Finally, our study confirms the multiple functions served by flavonoids, both flavonols and flavanols, in the adaptive mechanisms of plants to the environmental pressures associated to Mediterranean climate.

## 1. Introduction

There is great interest in understanding the mechanisms adopted by plants to cope successfully against the environmental challenges imposed by the Mediterranean climate [1]. The Mediterranean basin is a biodiversity hotspot, indeed, as it hosts about 16% of plant species worldwide while accounting for just 2% of land area [2,3]. The detrimental effects associated to unfavorable climate may be particularly severe in Southern Mediterranean, and especially during the summer season, when extended periods of elevated temperatures occur in concomitance with the scarcity of rainfall [4,5,6]. The challenges imposed by drought stress may be critical for the performance and the survival of plants inhabiting sunny areas, as solar irradiance dramatically exceeds the plant capacity to use it for carbon assimilation on both seasonal and daily basis [4,7,8,9,10]. Consequently, plants inhabiting harsh Mediterranean areas may transiently suffer from severe photo-oxidative stress [4,11,12], the intensity of which is predicted to increase further in the near future, because of climate change. It is not surprising, consequently, that hundreds experiments, performed at very different scale levels, have explored the responses of Mediterranean plants to drought stress (Mediterranean and drought scores 4207 documents in Scopus to date).

In contrast, the responses of Mediterranean plants to contrasting solar irradiance have been explored in less detail, particularly under in field conditions [13,14,15,16]. However, how plants adjust the suite of morpho-anatomical, physiological and biochemical traits in response to contrasting solar irradiance is long known [17,18,19,20,21]. Plants growing in full sunlight display leaves with reduced size, greater thickness and mesophyll density, and have usually steeper leaf angles compared to leaves inhabiting shaded areas [22,23]. Sunny plants are bushy in their nature, whereas shaded plants display long internodes and pronounced apical dominance, thereby sustaining either self-shading or maximum light interception, respectively, when plants face excessive or limited solar irradiance [13,24,25,26,27,28]. Major biochemical adjustments responsible for the successful acclimation to shade or full sun conditions involve both photosynthetic and non-photosynthetic pigments [29,30]. Usually, sun leaves have much lower concentration of chlorophyll on both leaf area and dry mass basis, substantially higher both Chl_a_/Chl_b_ and carotenoid (especially violaxanthin cycle pigments) to chlorophyll ratios than leaves growing in the shade [16,31,32,33]. This allows for sun leaves to both reduce centers of light absorption and dissipate excess energy into the chloroplast via nonphotochemical quenching (NPQ) [30,32,34]. Similarly, the concentration of non-photosynthetic pigments, such as the wide range of phenylpropanoid structures, is much higher in sunny compared to shaded leaves [13,35,36,37]. This equips leaves with an effective shield against the most energetic solar wavelengths and, at the same time, offers greater capacity to scavenge reactive oxygen species (ROS) generated by the unbalance between the rate of electrons available and the capacity to use it for carbon assimilation [29,37]

However, molecular events that are the base of the strategies adopted by plants, particularly those inhabiting the Mediterranean areas, to cope with the challenges imposed by limited or excessive solar radiation have been scarcely explored [21,38,39,40]. The matter is of great significance, as may help to explain mechanistically the profound physiological and biochemical adjustments plants activate to acclimate successfully to shade or full sunlight, when concomitantly challenged by other environmental stressors. In our study, we took advance of next generation sequencing technique (RNA-Seq) and targeted metabolomics to explore the mechanisms responsible for the acclimation of *Cistus incanus* in the shade (daily solar irradiance less than 10% of full sunlight irradiance) or in full sunlight (plants growing on seashore dunes). *C. incanus*, as most *Cistus* spp. plays a key role in ecosystem services in most degraded Mediterranean areas [41,42,43]. Though distributed in areas largely differing in light availability, *C. incanus* represents a key member of vegetation growing on seashore dunes. As a result, *C. incanus* is severely challenged not only against excessive solar radiation, but also by the detrimental effects induced by both the scarcity of water and the elevated temperatures, during the long summer season [42,44]. In our study, we performed (1) measurements of relevant physiological-related traits, such as leaf water potential, gas exchange and PSII photochemistry; (2) de-novo assembly of *C. incanus* transcriptome using RNA-Seq technology and the analysis of differentially expressed genes; (3) the identification and quantification photosynthetic pigments, abscisic acid, and flavonoids. To our knowledge, this is the first report dissecting the molecular mechanisms regulating the acclimation to shade or full sun conditions of Mediterranean plants.

## 2. Results

### 2.1. Water Relations, Gas Exchange and PSII Photochemistry

Shade and sun plants largely differed for both leaf thickness and leaf mass per area (LMA) which were much higher (on average +80%) in sun compared to shade leaves. Shade as well as sun plants suffered from severe drought stress, since leaf water potential was low even at predawn (ψ_w_^PD^ was on average −2.54 MPa), not only during the hottest hours of the day (ψ_w_^MD^ was on average −3.24 MPa) (Table 1). Leaves growing in the shade or in full sunlight displayed similar, very low CO_2_ assimilation rates, whereas stomatal conductance (*g*_s_) and intercellular CO_2_ (*C*_i_) concentration were 47% and 17% higher, respectively, in sun than in shade leaves. These findings are consistent with the observation that PSII photochemistry was impaired more in sun than in shade leaves, as revealed by both maximal (F_v_/F_m_) and particularly actual quantum yield (Φ_PSII_) of PSII photochemistry (Table 1). Excess excitation energy to the photosynthetic apparatus, as estimated by the chlorophyll fluorescence-derived parameter 1 – *q*_P_ (where *q*_P_ is the photochemical quenching) was consistently much higher (+134%) in sun than in shade leaves, as also observed for the leaf ability to thermally dissipate excess energy through nonphotochemical quenching (NPQ, +66%). We also observed that in sun leaves the ability to use available electrons for carbon assimilation was steeply lower compared to shade leaves, since the ratio of electron transport rate (ETR) to *A*_N_ in sun leaves exceeded by 143% that in shade leaves.

### 2.2. Transcriptome Analysis

In our study, we carried out the RNA-Seq of two cDNA libraries constructed from leaves of “shade plants” and “sun plants” of *C. incanus*, as detailed below in Materials and Methods. The reads, 18,769,137 and 18,218,889 for shaded and sun leaves, respectively, were *de novo* assembled to generate a collection of 59,030 transcripts with average length of 1028 bp and N50 at 1462 bp (Table 2). Assembled fasta sequences are publicly available on Figshare at: https://figshare.com/s/a6610b92dedaa72f84ea. Analysis of our transcriptome assembly identified 243 out of the 248 core proteins (98%) as ‘complete’ (defined as >70% alignment length with core protein) and 248 (100%) as ‘partial’. In addition, transcripts were subjected to BlastX (E-value < 1e^−5^) homology search against the plant UniProt database at the National Center for Biotechnology Information (NCBI, http://www.ncbi.nlm.nih.gov/) allowing annotation of 22,784 (38.6%) transcripts. The species distribution showed that the 22% had top match to the *Theobroma cacao* genes. To characterize further the biological pathways coverage of our transcriptome, all the assembled transcripts were mapped to the Kyoto Encyclopedia of Genes and Genomes (KEGG). Overall, 6760 transcripts were assigned to 359 KEGG pathways. Gene expression based on RNA-Seq data was estimated as RPKM (reads per kilo base of exon model per million mapped reads). A total of 3590 transcripts were identified as differentially expressed based on absolute fold change greater than 2 and *P*-value <0.05, with 2312 down-regulated and 1278 up-regulated transcripts in sun compared to shade *C. incanus* plants.

### 2.3. Analysis of Differentially Expressed Genes

Data reported in Figure 1 show large differences in the expression of a wide range of genes between shade- and sun-exposed leaves. These have been grouped into different sets based on functional analysis: ‘response to light’, ‘sucrose synthesis and transport’, ‘oxidative stress response’, Methyl Erythrytol Phosphate (MEP) and flavonoid biosynthetic pathways. A small set of transcription factors belonging to both the *WRKY* and the *zinc finger* family also differed between shade and sun leaves.

#### 2.3.1. Genes Involved in Light Response, Sucrose Synthesis and Transport, and Photooxidative Stress Protection

A set of genes involved in light perception and signaling were differentially expressed in shade and sun leaves (Figure 1 and Figure 2). In detail, non-phototropic hypocotyl 3 (*NPH3*), the cryptochrome-interacting basic-helix-loop-helix 1 (*CIB1*) and the Cryptochrome 2 (*CRY2*), the phytochrome interacting factor 3-like 6 (*PIL6*), and phytochrome kinase substrate 1 (*PKS1*) genes were overexpressed in shade leaves. Furthermore, leaves fully exposed to solar irradiance had significantly lower expression of genes involved in the assembly of both PSII (*D2*/*PsbD*; PSII *CP43, PsbC*) and PSI (Photosystem I P700 chlorophyll a apoprotein A1/A2), in the stability of PSII complex (*PsbP*), as well as in the effective light harvesting (PSII *CP47*). These results are in line with the observation that in sun leaves the expression levels of two ATP-dependent zinc metalloprotease, FTSH1 and FTSH2, which regulates the selective degradation of photodamaged D1 protein, were higher compared to shade leaves (Figure 1). While the expression of a gene coding for Ribulose-1,5-bisphosphate carboxylase/oxygenase (*Rubisco*) was significantly lower in sun compared to shade leaves, the reverse was detected for the expression of Rubisco activase (*RA*). 

The expressions of genes regulating the synthesis (sucrose phosphate synthase (*SPS1F*) and sucrose synthase (*SS*)) and the transport (*SUT1*, *SUT2*, and *SUT* genes involved in phloem loading) of sucrose were significantly higher in leaves growing in full sunlight. On the contrary, genes regulating sucrose efflux to the phloem apoplasm (driven by bidirectional sugar transporters *SWEET* genes), were either over-expressed or down regulated in sun leaves (Figure 1 and Figure 2).

In sun-exposed leaves the expressions of a suite of genes belonging to the family of Glutathione S-transferases (*GST*), which act as glutathione peroxidases and, hence involved in the detoxification of reactive oxygen species (ROS) were largely higher compared to shade-exposed leaves (Figure 1 and Figure 2), with the notable exception of a nuclear Tau class glutathione transferase, *GSTU45*. The expressions of other genes regulating the detoxification of reactive oxygen species (ROS), such as ascorbate peroxidase (*APX*) and dehydroascorbate reductase (*DHAR*), both involved in the detoxification of H_2_O_2_ and in the glutathione-dependent reduction of oxidized ascorbate, respectively, were also higher in sun compared to shade leaves. Similarly, the expression of Lactoylglutathione lyase (*LGL*), which regulate the detoxification of methylglyoxal, the concentration of which increases because of soluble carbohydrates accumulation and lipid peroxidation, was substantially higher in sun leaves.

#### 2.3.2. Genes Regulating the MEP and Flavonoid Biosynthetic Pathways

In our study, shade leaves displayed higher expression of a gene involved in the early steps of MEP pathway, i.e., 1-deoxy-D-xylulose-5-phosphate reductoisomerase (*DXR*), whereas the gene involved in the first committed step of carotenoid biosynthesis (phytoene synthase, *PSY*) was overexpressed in sun leaves (Figure 1 and Figure 3). Among the suite of genes involved in the synthesis of individual carotenoids, β-carotene hydroxylase (*CA*), which regulates the synthesis of zeaxanthin from β-carotene, was steeply overexpressed in sun leaves. Sun leaves also had higher expression of β-carotene isomerase, a cis-to-trans isomerase (also known as *DWARF27*), which regulates the synthesis of cis β-carotene, and subsequently the synthesis of strigolactones. More subtle results originate from the analysis of genes involved in the synthesis of abscisic acid. While the expression of zeaxanthin epoxidase (*ZEP*), which regulates the first committed step in ABA biosynthesis, was overexpressed in sun leaves, the expressions of two members of the nine-cis-epoxycarotenoid dioxygenase (*NCED*) gene family (*NCED4* and a not-fully annotated *NCED*), which regulate the biosynthesis of ABA from neoxanthin/violaxanthin, were downregulated in sun leaves (Figure 1 and Figure 3).

As expected, a suite of genes regulating early and late steps in the flavonoid biosynthetic pathway were largely overexpressed in sun leaves (Figure 1 and Figure 3). This included early genes, such as chalcone synthase (*CHS*), chalcone flavonone isomerase (*CHI*), flavanone 3-hydroxylase (*F3H*), flavonoid 3′- (*F3*′*H*) and flavonoid 3′-5′-hydroxylase (*F3*′,*5*′*H*), as well as late genes, such as flavonol synthase (*FLS*), dihydroflavonol reductase (*DFR*), leucoanthocyanidin reductase (LAR) and anthocyanidin reductase (*ANR*). This observation is consistent with the flavonoid composition of *C. incanus* leaves, in which derivatives of the flavonol quercetin and myricetin occurs in concomitance with large concentrations of gallocatechin and proanthocyanidin polymers derivatives [45]. We also observed that sun leaves had higher expression levels of genes coding for proteins involved in the transport of flavonoids, such as members of the multidrug resistance (*MDR*) and ATP-binding cassette (*ABC*) families.

#### 2.3.3. WRKY and Zing Finger Transcription Factors

Members of the *WRKY* and zinc finger families of TFs, long reported as being involved in the response of plants to stress agents of different origin (of both abiotic and biotic nature) were largely overexpressed in sun-exposed leaves (Figure 1).

### 2.4. Photosynthetic Pigments, Abscisic Acid and Flavonoids

While Chl_tot_ concentration was largely higher (+84%) in shade than in sun leaves, the concentration of total carotenoids (Car_tot_) in sun leaves slightly (+19%), but significantly exceeded that in shade leaves (Figure 4). Light-induced increase in Car_tot_ concentration was the result of an increase in the concentration of xanthophylls (+16%), particularly of violaxanthin-cycle pigments (VAZ, +44%). The concentration of VAZ relative to Chl_tot_ (VAZ Chl_tot_^−1^) increased as much as 166%, indeed, passing from shade to sun leaves. Light-induced enhancement in the de-epoxidation state (DES) of VAZ was observed, although to much less extent (+20%) than expected. In our study, significant difference in the concentrations of free-ABA and ABA-glucosyl ester (ABA-GE) were not observed between differentially irradiated leaves (Figure 4).

The leaf flavonoid concentration was affected by solar irradiance, since in sun-exposed leaves the total flavonoid concentration exceeded by 71% that detected in shade-exposed leaves (Figure 4). Major light-induced increases were observed for myricetin (+610%) and quercetin derivatives (+210%), whereas shade-to-sun increments in both catechin and pro-anthocyanidin polymers were less evident (+44%). We did not observe significant difference in kaempferol derivatives between the shade and sun leaves.

## 3. Discussion

Our study offers a comprehensive picture of the molecular events allowing *C. incanus* to acclimate at limited or excessive solar irradiance, when concomitantly suffering by drought stress during Mediterranean summer. In our study, sun-adapted plants suffered from more severe challenges imposed by the concomitant actions of water stress, high solar irradiance and elevated air temperatures compared to shade-adapted plants. Consequently, our discussion predominantly, but not exclusively focuses on the suite of integrated and modular events, operating at different levels of scale, allowing sun plants to cope with most severe environmental pressures.

First, our study reveals major differences in genes regulating both light perception and transduction. We have observed that a suite of genes primarily involved in the regulation of shade avoidance response/syndrome [46,47] was largely overexpressed in *C. incanus* (a sun-loving species [41,42]) when growing in full shade [48]. This includes *NPH3* and *PKS1*, which act in concert, and downstream of the blue photoreceptor NPH1, in promoting leaf flattening and highly oriented growth [49,50,51], as well as *PIL6*, whose overexpression is known to promote hypocotyl elongation [52]. Consistently, the expressions of two relevant members of the cryptochrome family (*CRY1* and *CRY2*), which are known as modulating the activity of phytochrome interacting factors *PIF4* and *PIF5* under low blue-light irradiance, were also largely higher in shade compared to sun leaves [47,53,54,55]. In contrast, sun leaves exhibited much higher expression of β-carotene isomerase (*DWARF27*), which regulates the first step of strigolactones biosynthesis, thus promoting shoot branching [56]. Our study also reveals that the expression of a member of *CONSTANS* zing finger family of TFs, possibly contributed to shape the canopy architecture in sun-exposed plants, through the regulation of auxin gradients, as usually observed under high red to far red ratio [57,58]. 

Second, we have shown that the combined action of water deficit and elevated temperatures (i.e., drought stress *sensu stricto*) was much likely responsible for the large impairment of PSII photochemistry (and photosynthesis) in plants growing in full sunlight, as previously reported in a range of species [59,60,61,62]. In our study, leaf water potential was lower indeed, whereas leaf temperature was substantially higher in sun (midday leaf T averaged 34.9 °C) than in shade (leaf T was on average 31.8 °C) leaves. Though the observed differences in leaf T between shade and sun leaves may have a relatively minor impact, *per se*, on leaf photosynthetic performance, such increments in leaf T may severely limit PSII photochemistry when coupled with water stress [11,63].

Our study also reveals that the expressions of genes regulating the assembly, the stability and the effective functioning of both PSII (*PsbB*, *PsbC*, *PsbD*, *PsbP*) and PSI (*PsaA*, *PsaB*) were downregulated in sun leaves, consistent with the steeply lower PSII quantum yield in the dark and, particularly in the light-adapted state, displayed by sun compared to shade leaves [64,65]. Major detrimental effects of Mediterranean summer on PSII functioning in sun-growing *C. incanus* plants are additionally revealed by the extent to which PSII suffered from excess excitation energy (1 − *q*_P_) and from the much lower ability of sun compared to shade leaves in using available electrons for carbon fixation (ETR/*A*_N_). This is in line with observation that so-called PSII repair cycle, which is crucial for plant survival under severe light excess, and sustained by *FTSH1* and *FTSH2* was more active in sun- than in shade-adapted leaves [66,67]. FTSH1/2 are ATP-dependent zinc metalloprotease that sustain the rapid degradation of D1 damaged protein, thereby promoting *de-novo* D1 synthesis and the partial reassembly of PSII into grana thylakoid [68]. 

Overall, these findings conform to the observation that non-stomatal limitations contributed more in sun than in shade leaves to constrain net CO_2_ assimilation rate (C*i* was indeed 27% higher in sun than in shade leaves). In our study, genes coding for Rubisco were largely downregulated in sun leaves, and only in part compensated by an increase in the expression of Rubisco activase compared to shade leaves. We hypothesize that in sun leaves sugar phosphates accumulated to greater degree to shade leaves, based on the large overexpression of sucrose phosphate synthase genes, and that Rubisco activase was unable to effectively remove sugar phosphates from the Rubisco active sites in sun leaves [69,70,71]. Data of our study agree with previous findings of a major role of leaf T, not only of light irradiance, in determining the optimal ratio of Rubisco activase to Rubisco, and hence the whole-leaf carboxylation efficiency [71,72].

Third, our study reveals additional events which operate at the organism and whole-plant levels that may have been responsible for the steeply lower ability of sun leaves to use available radiant energy for carbon fixation. In our study, photosynthetic radiation use efficiency [73], varied from 0.0064 in shade leaves to 0.00072 in sun leaves, thereby translating into severe excess of reducing power in sun compared to shade leaves. This is also consistent with ETR/*A*_N_ values observed in our experiment, which were indeed much higher (+131%) in sun than in shade leaves. It is possible that ETR in sun leaves has been overestimated in our study, in which an equal distribution of photons between PSII and PSI has been assumed. There is evidence that the portion of light energy distributed to PSII may decrease upon long exposure to high solar irradiance [74]. Nonetheless, such a decline in PSII-to-PSI photon distribution in severely high light stressed leaves should have only in part contributed to the large changes in ETR between shade and sun leaves, observed in our study.

Furthermore, we speculate that higher sugar (and sugar intermediate) biosynthesis in sun compared to shade leaves should have partially countered the steeper decline in leaf ψ_w_ detected in sun leaves, since leaf bulk osmotic potential (ψ_π_) at midday was significantly lower (−4.15 MPa) in sun than in shade leaves (−3.65 MPa). This should have also triggered feedback downregulation of photosynthesis [75,76,77], since a parallel limitation in the export of sucrose to the phloem likely occurred in sun-adapted leaves. At the whole-plant level, our data are also consistent with a decreased export of carbohydrates, because of decreased strength of sink tissues [78] in the semi-deciduous *C. incanus*, the growth of which is negligible during the summer season [41].

Fourth, as expected, in sun leaves a suite of antioxidant defenses were activated to counter the severe oxidative stress sun leaves should have experienced during the summer season. This included a wide range of Glutathione S-Transferases (GSTU) that may operate both in the chloroplast (i.e., GSTF8) and possibly in other cellular compartments [79], including the cytosol, to effectively remove high light-induced singlet oxygen and H_2_O_2_ generation [80,81]. The increased ability of sun leaves compared to the shade-adapted counterparts in H_2_O_2_ removal is also consistent with the observation that both *APX* and the companion *DAR*, which is indeed involved in the glutathione-mediated recycling of oxidized ascorbate, were largely overexpressed in sun leaves [82,83]. Consistently, the expressions of a range of *TFs*, previously reported, as being involved in light stress-induced ROS generation, was higher in sun than in shade leaves. These included both *WRKY46* and *WRKY70*, [84,85], as well as the zinc finger *GroES* gene (also known as chaperonin 10), which is responsive to heat stress and involved in reactive species homeostasis, including NO detoxification [86]. Moreover, and consistent with the high expression levels of genes involved in the biosynthesis of sucrose and sucrose intermediates, a gene coding for lactoylglutathione-lyase, which detoxify methylglyoxal, a toxic product generated by both high sugar accumulation was largely overexpressed in sun leaves [87,88]. 

Finally, our study offers compelling evidence of the profound metabolic reprogramming within the MEP and flavonoid pathways, aimed not only at limiting the flux of solar irradiance, but also at countering the photooxidative damage generated in sun leaves by the concomitant action of high solar irradiance, water deficit and elevated temperatures [89]. We have indeed observed that de-epoxidation of the VAZ pool was greater, to sustain the superior need to dissipate thermally excess energy through nonphotochemical quenching (NPQ) in sun-exposed compared to the shade-exposed counterparts. Nonetheless, in our study, the high relative (to Chl_tot_) concentration of VAZ suggests these xanthophylls only in part contributed to NPQ, since their concentration was high enough to saturate the potential binding sites in antenna proteins [30,90]. The analysis of DEG in our study, offers strong support to the previous suggestion that light-induced increases in zeaxanthin originated mostly through hydroxylation of β-carotene [91,92]. This is consistent with the observation that an increase in the expression of β-carotene hydroxylase gene was accompanied by a parallel increase in the expression of the zeaxanthin epoxidase passing from shade- to sun-adapted leaves. The synthesis of zeaxanthin from β-carotene occurs in response to severe drought stress [6,91,92], as was the case of sun-exposed plants in our study. Therefore, this pool ‘free’ zeaxanthin located in the lipid phase of thylakoids, mostly contributed to increase membrane rigidity through stabilization of phospholipids layers, and limiting membrane lipid peroxidation, rather than sustaining thermal dissipation of excess energy through NPQ [6,93,94,95]. In other words, zeaxanthin behaved primarily as a chloroplast antioxidant in severely stressed sun-adapted *C. incanus* leaves. 

Zeaxanthin-epoxidase also regulates the first committed step of ABA biosynthesis. However, in sun leaves, the higher expression of *ZEP* coincided with the dramatically lower expressions of two members of the nine-cis-epoxycarotenoid dioxygenase (*NCED*) family, as also observed in *Fagus sylvatica* [96]. This may in part explain similar concentrations of free- and conjugated ABA detected in shade and sun leaves. Since a large fraction of DEG has not been successfully annotated, we cannot exclude that genes involved in both the release of free-ABA from ABA-GE and the catabolic oxidation of ABA, both highly responsive to light irradiance [93,97,98,99], may have determined the levels of foliar ABA observed in shade or-sun-adapted leaves, in our study. Foliar ABA levels did not match stomatal conductance (g_s_ was higher in sun than in shade leaves [100,101]. We hypothesize that hydraulic signals or ABA in the xylem sap, more than whole-leaf ABA levels, might have contributed to regulate stomata movements, although we cannot exclude that just a fraction of foliar ABA is actually involved in the biochemical control of stomata aperture, as previously reported in drought-stressed tobacco [102]. 

Our study highlights the key contribution of flavonoids, including flavonol and flavanol (flavan 3-ol) derivatives, in the acclimation of *C. incanus* to high sunlight. A suite of (early – *CHS*, *CHI*, *F3H*, *F3’H*, *F3’,5’H* -, and late – *FLS*, *DFR*, *LAR*, *ANS* -) genes regulating flavonoid biosynthesis and transport, (including MDR, ABC and GST proteins) [37,103] was largely overexpressed in sun-adapted leaves. The parallel increases in the expression of genes involved in flavonoid biosynthesis and transport out of the endoplasmic reticulum reveals a highly coordinated system sustaining the effective delivery of metabolites to different subcellular sink organs [104,105,106,107]. Light-induced preferential accumulation of quercetin and myricetin glycosides, coupled with negligible variation in kaempferol derivatives, strongly supports the idea that flavonols serve the dual role of UV-screeners and antioxidants in sun-adapted leaves [29]. Further contribution to superior capacity to screen off the most energetic solar wavelengths in sun- compared to shade-exposed leaves comes from the light-induced increased concentration of flavanol derivatives, which are effective in absorbing wavelengths over the 280–310 nm region of the solar spectrum. In our study, condensed tannins might have served another key role in sun-exposed plants, which additionally suffered from severe drought stress. Bussotti et al. [108] have reported large inter-cellular redistribution of tannins on seasonal basis, with tannins impregnating the outer wall of epidermal cells, thereby both contributing to cell lignification and reducing cuticular transpiration, in leaves suffering from severe water deficit. Though ultrastructural analysis has not been conducted in our study, it is plausible that intra- and inter-cellular re-distribution of tannins might have occurred to greater degree in sun-adapted compared to shade-adapted leaves, as revealed by the expression levels of genes involved in flavonoid transport. This may have also contributed to the markedly higher sclerophyll index (leaf mass per area, LMA) and leaf thickness of sun compared to shade leaves.

## 4. Materials and Methods 

### 4.1. Plant Material and Growth Conditions

The experiment was conducted on *Cistus incanus* L. plants growing either under a dense overstory of *Pinus pinea* L. (shade plants) or on seashore dunes (sun plants) at Castiglione della Pescaia (42° 46′ N, 10°53′ E). Total integrated photon flux densities were monitored on several clear days before and after the sampling date, July 22nd, with a 1800 LI-COR spectroradiometer (LI-COR, Lincoln, NE, USA) and averaged 3.88 and 58.2 mol·m^−2^·d^−1^ for shade and sun sites, respectively. Photosynthetic active radiation (over the 400–700 nm waveband) at midday, averaged 1970 or 138 μmol photons m^−2^·s^−1^ at the at the sun or shade site, respectively. Maximum air temperature, which was recorded at the “Ponti di Badia” weather station (located 4 km from the study site), averaged 31.7 ± 1.8 °C, during the experiment. All non-destructive measurements as well as leaf sampling for biochemical analyses were conducted during the midday hours, from 12:00 to 14:00 h, on fully developed (third to fourth node from leaf apex) leaves. 

### 4.2. Leaf Water Potential, Gas Exchange and PSII Photochemistry

Leaf water potential, gas exchange and PSII performance were determined using protocols previously reported in Tattini et al. [6,36]. Leaf water potential (Ψ_w_) was measured using a Scholander-type pressure chamber (PMS Instruments, Corvallis, OR, USA). Net photosynthesis (*A*_N_) and stomatal conductance (g_s_) were measured with a LI-6400 portable photosynthesis system (Li-Cor, Lincoln, NE, USA) and inside the cuvette, temperature, leaf-to-air vapor pressure deficit and CO_2_ flow rate were kept at 28 ± 1.3 °C, < 1 kPa and 300 μmol·mol^−1^, respectively. The leaf was left equilibrate inside the chamber for 10 min at ambient CO_2_ concentration and saturating irradiance (which was set at 200 or 1800 μmol m^−2^·s^−1^, for shade and sun leaves, respectively) before measurements.

Chlorophyll fluorescence analysis was performed using a portable PAM-2000 Chl fluorometer (Heinz Walz, Effeltrich, Germany) on dark-adapted leaves (over a 30-min period). Minimum fluorescence (F_0_) was measured with a weak, modulated measuring beam of 0.8 μmol quanta m^−2^·s^−1^. Before measuring light-adapted fluorescence parameters, fluorescence-measuring light was switched-off for 2 min, before illuminating the leaves with actinic light (200 or 1800 μmol quanta m^−2^·s^−1^, for shade or sun leaves, respectively). To determine light-adapted maximal fluorescence (F_m_′), 10 pulses of saturating white light (800 ms) were applied at 20-s intervals during actinic illumination. After the saturating pulse, maximal fluorescence reached F_m_′ value, and actinic light allowed steady-state photosynthesis and modulated fluorescence yield to be reached (F_s_). Fluorescence induction kinetics was monitored and different parameters determined when stationary. Light-adapted initial fluorescence (F_0_′) was determined during a brief interruption of actinic illumination in the presence of far-red illumination, thereby allowing preferential excitation of PSI [109]. Under this condition, QA (the primary quinone acceptor in photosystem II (PSII)) is assumed to become rapidly photo-oxidized with consequent conversion of PSII centers into the open state. These data were used to calculate nonphotochemical quenching (NPQ = (F_m_ – F_m_’)/F_m_’) [110], and the actual quantum yield of photosystem II (ΦPSII = (F_m_’ − F_s_)/F_m_’ [111]. The electron transport rate (ETR) was calculated as ETR = 0.5 × Φ_PSII_ × PPFD × leaf absorptance (α). This equation assumes an equal distribution of photons between PSII and PSI (hence the factor 0.5), which might led, however, to overestimate ETR in sun-exposed leaves (as detailed in Section 3, Discussion).

Leaf absorptance of 0.84 and 0.88 for shade and sun leaves, respectively, was determined using a LI-COR 1800-125 integrating sphere coupled to a LICOR 1800 spectroradiometer, as reported in Tattini et al. [36]. The absorptance over the 400–700 nm waveband was calculated as α = 1 – Reflectance – Transmittance. Finally, whole leaf thickness and leaf mass per area were determined using the protocols reported in Tattini et al. [36].

### 4.3. RNA-Seq Analysis, Library Preparation and Transcriptome Sequencing

Leaf samples were immediately frozen in liquid nitrogen and stored at −80°C prior to analysis. Highly pure total RNA was extracted using RNeasy Plant Mini Kit (Qiagen, Valencia, CA, USA) according to manufacturer’s instructions with little modifications as detailed in Torre et al. [112]. Purity and quantity of each RNA sample were checked through spectrophotometry and gel electrophoresis. The final quality assessment was performed using an Agilent 2100 Bioanalyzer (Agilent Technologies, Santa Clara, CA, USA), with a minimum accepted RNA integrated number (RIN) of 8. Two RNA-Seq libraries, corresponding to sun and shaded *C. incanus* leaf samples were prepared using Illumina TruSeq RNA sample Prep Kit (Illumina, Inc., CA, USA). cDNA libraries construction and paired-end (2 × 100) sequencing, using Illumina HiSeq2000 system (Illumina Inc.) was performed at IGA Technology Services S.r.l. (Udine, Italy). The raw reads have been deposited at NCBI’s Sequence Read Archive (SRA) database with the following accession information: Bioproject ID: PRJNA354245; Biosamples: shade leaves, SAMN06044962 (SRA: SRR5040963); sun leaves, SAMN06044963 (SRA: SRR5040964).

### 4.4. Transcriptome de novo Assembly and Annotation

The Illumina reads were processed to remove low quality ends (typically N, maximum value = 2) using ERNE-FILTER (www.erne.sourceforge.net), using default parameters, with the exception of minimum-size and errors-rate, which were fixed at 50 and 25, respectively. Sequence reads were processed further for quality assessment with FastQC [113]. In both libraries we maintained a phred-like quality score (*Q*-score) > 20 for downstream analysis (average per base quality score = 37). The reads, were combined and de novo assembled using Trinity v. 20130225 [114] with default k-mer size of 25. Transcripts longer than 200 bp were selected and clustered at 95% identity using CD-HIT-EST v4.6.1 [115]. In addition, contigs with low support (fpkm cutoff = 1.0) were filtered from the *Cistus incanus* transcriptome assembly. Reads abundance for each contig was estimated by mapping back the reads used for this assembly to the contigs using Bowtie program [116] and, subsequently, the RSEM algorithm [117]. Assembly completeness was estimated through identification of conserved core eukaryotic proteins with CEGMA analysis [118]. Similarity search of assembly against a subset of KOG database consisting of 248 highly conserved proteins, from a wide range of eukaryotes, was conducted.

### 4.5. Analysis of Differentially Expressed Genes

Gene expression levels were calculated using the reads per Kilobase per million mapped reads (RPKM) method [119], by mapping raw reads of green and red basil to the basil reference transcriptome. The reference transcriptome of sweet basil was assembled using the Illumina platform. Sequence reads were deposited at the Sequence Read Archive (SRA) of the National Center of Biotechnology Information (NCBI, Bethesda, MD, USA) under accession number SRA313233.

The relative expression of 11 selected genes was measured through qRT-PCR to confirm the DEGs highlighted by RNA-Seq. Primer pairs (Appendix A) were designed based on newly assembled transcripts by means of Primer3 software [120]. Total RNA was extracted from *C. incanus* leaves as already described above and reverse-transcribed by using SuperScript VILO cDNA Synthesis Kit (Invitrogen, Carlsbad, CA, USA). qRT-PCR was carried out in a StepOnePlus real-time PCR system (Applied Biosystems, Foster City, CA, USA) with SYBR green technologies (Power SYBR green PCR Master Mix; Applied Biosystems) according to the manufacturer’s instruction. Cycling conditions were: 95 °C for 2 min, 40 cycles at 95 °C for 10 s, 60 °C for 15 s, and 72 °C for 15 s. Measurements were performed on three replicates and the products were verified by melting curve analysis. The relative expression levels of the selected unigenes normalized to β-Actin were calculated using the 2^−ΔΔCt^ method [121].

### 4.6. Analysis of Photosynthetic and Non-Photosynthetic Pigments, and Abscisic Acid

Individual carotenoids and chlorophylls were identified and quantified as reported in Tattini et al. [102]. Freeze-dried leaf material (200 mg) was extracted with 2 × 5 mL acetone (added with 0.5 g·L^−1^ CaCO_3_) and 15 µL of supernatant injected into a Perkin Elmer Flexar liquid chromatograph equipped with a quaternary 200Q/410 pump and an LC 200 diode array detector (all from Perkin Elmer, Bradford, CT, USA). Photosynthetic pigments were separated in a 250 × 4.6 mm Agilent Zorbax SB-C18 (5 µm) column operating at 30 °C, eluted for 18 min with a linear gradient solvent system, at a flow rate of 1 mL·min^−1^, from 100% acetonitrile–methanol (95/5 with 0.05% of triethylamine) to 100% methanol–ethyl acetate (6.8/3.2). Violaxanthin cycle pigments (violaxanthin, antheraxanthin, and zeaxanthin, collectively named VAZ), neoxanthin, lutein, and β-carotene were identified using visible spectral characteristics and retention times. Individual carotenoids and chlorophylls were calibrated using authentic standards from Extrasynthese (Lyon-Nord, Genay, France) and Sigma-Aldrich (Milan, Italy), respectively.

The concentrations of free ABA and ABA glucoside (ABA-GE) were determined in 200 mg of freeze-dried leaf tissue, ground in liquid nitrogen, to which 50 ng of deuterated abscisic acid (d_6_-ABA) and 50 ng of deuterated ABA-GE (d5-ABA-GE, from the National Research Council of Canada, Saskatoon, SK, Canada) were added. The tissue was extracted with 3 mL methanol-water (50/50 adjusted to pH 2.5 with HCOOH) for 30 min at 4 °C, and the supernatant partitioned with 3 × 3 mL n-hexane. The methanol-water fraction was loaded onto Sep-Pak C18 cartridges (Waters, Milford, MA, USA), which were washed with 2 mL pH 2.5 water, and then eluted with 1.2 mL ethyl acetate. The eluate, dried under nitrogen, was rinsed with 500 µL pH 2.5 methanol-water. Aliquots of 3 µL were injected into a liquid chromatography–electrospray ionization (ESI) tandem mass spectrometry (MS-MS) device, consisting of an Agilent LC1200 chromatograph coupled with an Agilent 6410 triple quadrupole MS detector equipped with an ESI source (all from Agilent Technologies, Santa Clara, CA, USA), operating in negative ion mode. Free ABA and ABA-GE were separated in an Agilent Poroshell C18 column (3.0 × 100 mm, 2.7 µm), eluted with a linear gradient solvent, at a flow rate of 0.3 mL·min^−1^, from 95% H_2_O (with 0.1% of HCOOH, solvent A) to 100% acetonitrile–methanol (50/50, with 0.1% of HCOOH, solvent B) over a 30-min run. Quantification of free ABA and ABA-GE was conducted in multiple reaction mode (MRM) as reported in López-Carbonell et al. [122].

Phenylpropanoids were identified and quantified following the protocol reported in Gori et al. [45]. Freeze-dried leaf tissue (300 mg) was extracted with 3 × 6 mL 70% ethanol adjusted to pH 2 with formic acid (HCOOH). The supernatant was partitioned with 3 × 5 mL n-hexane, reduced to dryness under vacuum and rinsed with methanol-water (50/50, pH 2). Identification of individual phenolics was carried out using their retention times and both UV–VIS, MS and MS/MS spectra. The LC–DAD-MS/MS system consisted of a Shimadzu LCMS-8030 quadrupole mass spectrometer (Kyoto, Japan) operating in electrospray ionization (ESI) mode and a Shimadzu Nexera HPLC system (Kyoto, Japan) equipped with a diode array detector (DAD), a degasser, two eluent pumps, a column oven and an auto-sampler. The separation was performed on a reversed-phase Waters Nova-Pak C18 column (4.9 × 250 mm, 4 µm), (Water Milford, MA, USA). The mobile phase consisted of 1% aqueous formic acid (solvent A) and 1% formic acid in acetonitrile/methanol (25/75) (solvent B). Separation was obtained using the following elution gradient: 2% B isocratic for 10 min, from 2% to 98% B linear for 30 min, 98% B isocratic for 7 min. The flow rate was 0.6 mL·min^−1^, and the injection volume was 10 µL. The column oven was set at 30 °C. The mass spectral data were acquired in the following ESI inlet conditions: nebulizing gas and drying gas was nitrogen at a flow rate of 3.0 and 15.0 L·min^−1^, respectively; the interface voltage was set to 3.5 kV; desolvation line (DL) temperature was 250 °C and the heat block temperature was 400 °C. The mass spectrometer operated in Negative Ion Scan and in Product Ion Scan mode using analyte-specific precursor ions, with Argon as CID (Collision Induced Dissociation) gas at a pressure of 230 kPa. Quantification of individual compounds was directly performed by HPLC–DAD in triplicates. In particular, six individual compounds, i.e., gallic acid, epicatechin, myricetin 3-O-rhamnoside, quercetin 3-O-rhamnoside, rutin, were quantified with their own standard curves. Calibration of epicatechin, myricetin and kaempferol derivatives was performed at 280 and 350 nm using epicatechin, myricetin 3-O-rhamnoside and kaempferol 7-O-glucoside as reference compounds, respectively. Quantification of proanthocyanidin polymers was performed based on the epicatechin calibration curve [109].

### 4.7. Statistics

The experimental design was at full random, with four replicates (four plants) per each light irradiance. Data (means ± SD, *n* = 4) were analyzed with one-way analysis of variance (one-way ANOVA) and significant differences among means were estimated at the 5% (*p* < 0.05) level, using Tukey’s test. RNA-seq was performed on six plants (three leaves per plant) per light irradiance. Data were analyzed with the *Z*-test [123] with |fold change| > 2 and adjusted *p*-value < 0.05, in order to identify genes with significantly different expression between shade and sun leaves (CLC Genomic Workbench, Qiagen, Hilden, Germany). The qRT-PCR analysis was performed in triplicate, and data are reported as mean ± SD.

## 5. Conclusions

Our study offer a comprehensive picture of the wide range of adjustments, which operate at different levels of scale, plants activate to acclimate/adapt to contrasting sunlight irradiance. We have shown here major mechanisms regulating light perception and signaling, particularly aimed at sustaining the shade avoidance syndrome in a ‘sun loving’ species. On the other side, adaptive mechanisms to high solar irradiance have been dissected in our study at the level of whole plant morphology, since the expression of genes regulating shoot branching, through both strigolactone biosynthesis (*DWARF27*) and regulation of auxin gradients (*CONSTANS*), were largely overexpressed in sunny leaves. 

We have also documented, not only the detrimental effects of high solar irradiance on the assembly and stability of PSII which, in turn, have resulted into the low use of photosynthetic radiation for carbon assimilation, but also the activation of intrinsic repair mechanisms in sun-exposed leaves. Our study evidences the multiplicity of events limiting carbon assimilation in severely drought-stressed sunny leaves, which include not only the limitation of carboxylation efficiency, but also the over-accumulation of soluble carbohydrates and the consequent feedback downregulation of net CO_2_ assimilation. Consistently, our study also reveals the suite of genes involved in the regulation of ROS homeostasis, which include not only *GSTs* and *APX*, but also a range of *TFs* (*WRKY* and *zinc finger* families). 

The profound light-induced cellular re-programming has been documented in depth in our study, as revealed by the analysis of genes regulating key steps of the MEP pathway, which is highly responsive to light irradiance. Here we have offered compelling evidence that (1) the increase in zeaxanthin level because of sunlight originated from hydroxylation of β-carotene and not from de-epoxidation of violaxanthin, consistent with sunny leaves having suffered from severe drought stress; and (2) zeaxanthin mostly contributed to increase membrane rigidity and limiting membrane lipid peroxidation, rather than sustaining NPQ. 

Our study does not offer conclusive evidence of light regulation of ABA biosynthesis, but stimulates further studies exploring the contribution of deconjugation of ABA-GE as well as of ABA catabolism in regulating the levels of foliar ABA in leaves suffering from multiple environmental pressures associated to Mediterranean climate. 

Finally, our study confirms a central role of flavonoids, both flavonols and flavanols, in the adaptive mechanisms of plants to excessive light, and offers the hypothesis of multiple functions of condensed tannins in high light-grown plants concomitantly challenged against water deficit, the significance of which deserves further investigation.

## Figures and Tables

**Figure 1 ijms-20-03599-f001:**
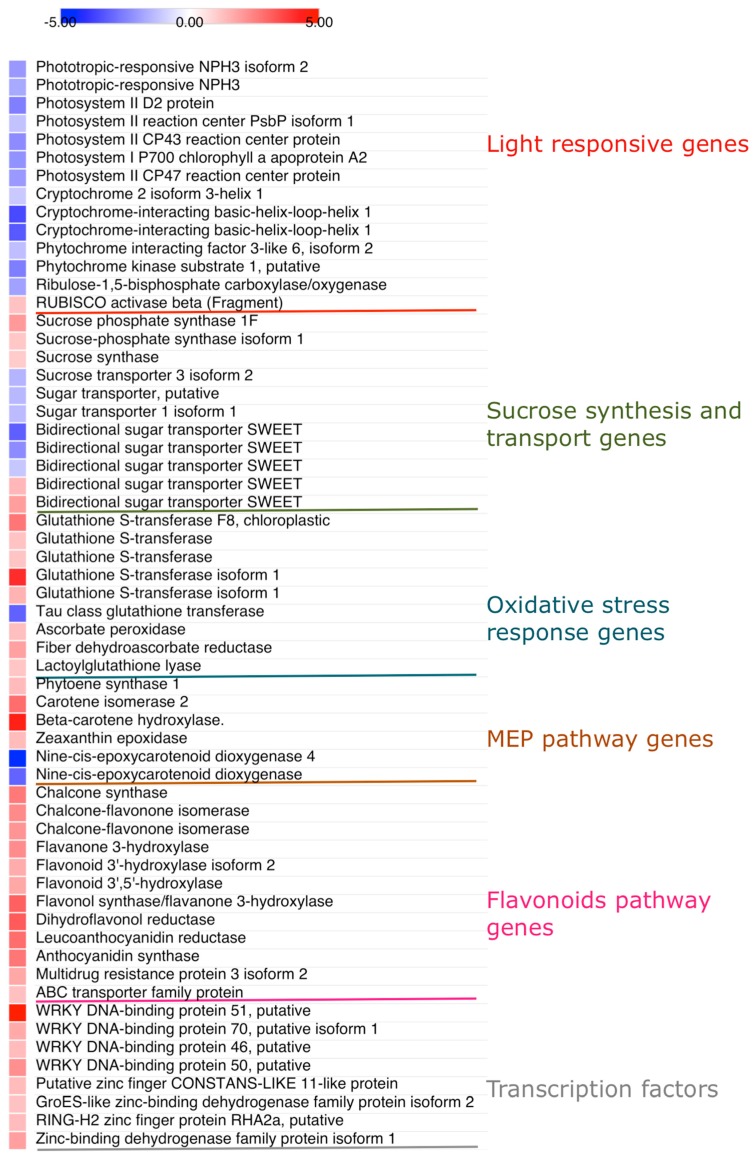
Heat map showing differentially expressed genes in *C. incanus* leaves adapted to shade or full sunlight. Fold changes, on log_2_ basis, denotes sun to shade transcript abundance.

**Figure 2 ijms-20-03599-f002:**
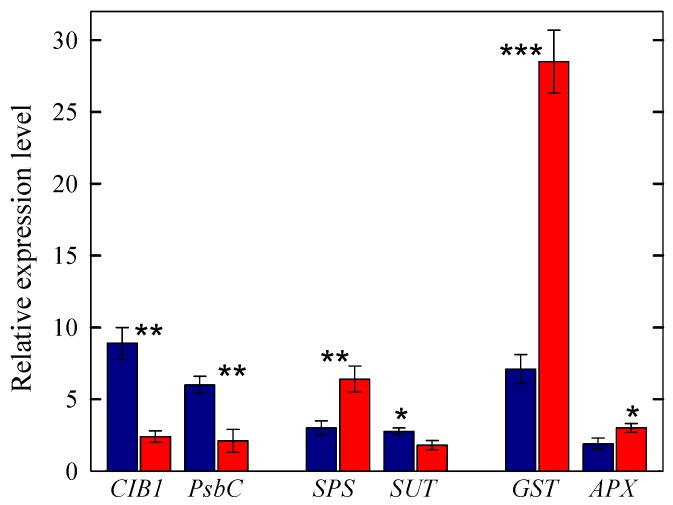
Relative expression levels of selected genes regulating light response (*CIB1*, *PsbC*), the synthesis and transport of sugars (*SPS*, *SUT*), and the oxidative stress response (*GST*, *APX*) in shade (blue bars) or sun (red bars) leaves of *C. incanus*. Data are means ± standard deviation (*n* = 3). Asterisks denote significant difference: *** *p* < 0.001; ** *p* < 0.01, * *p* < 0.05.

**Figure 3 ijms-20-03599-f003:**
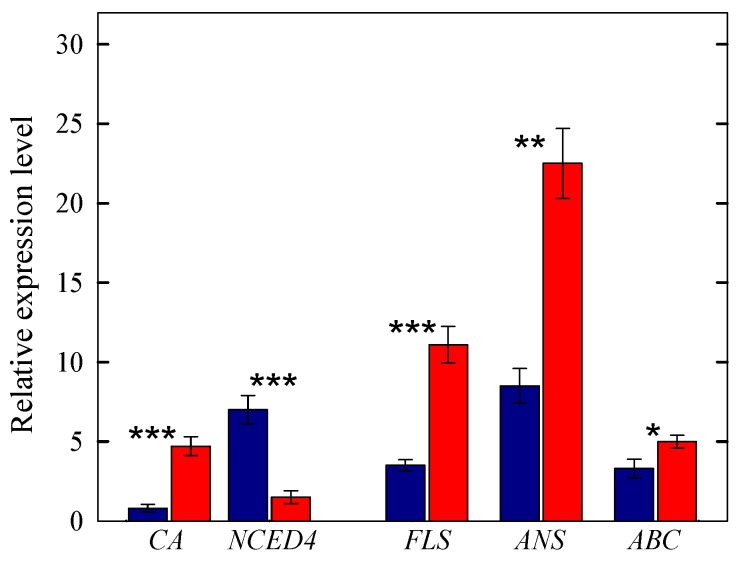
Relative expression levels of selected genes regulating the synthesis of MEP- (*CA*, *NCED4*) and flavonoid-derived products (*FLS*, *ANS*), as well as the transport of flavonoids (*ABC*) in shade (blue bars) or sun (red bars) leaves of *C. incanus*. Data are means ± standard deviation (*n* = 3). Asterisks denote significant difference: *** *p* < 0.001; ** *p* < 0.01, * *p* < 0.05.

**Figure 4 ijms-20-03599-f004:**
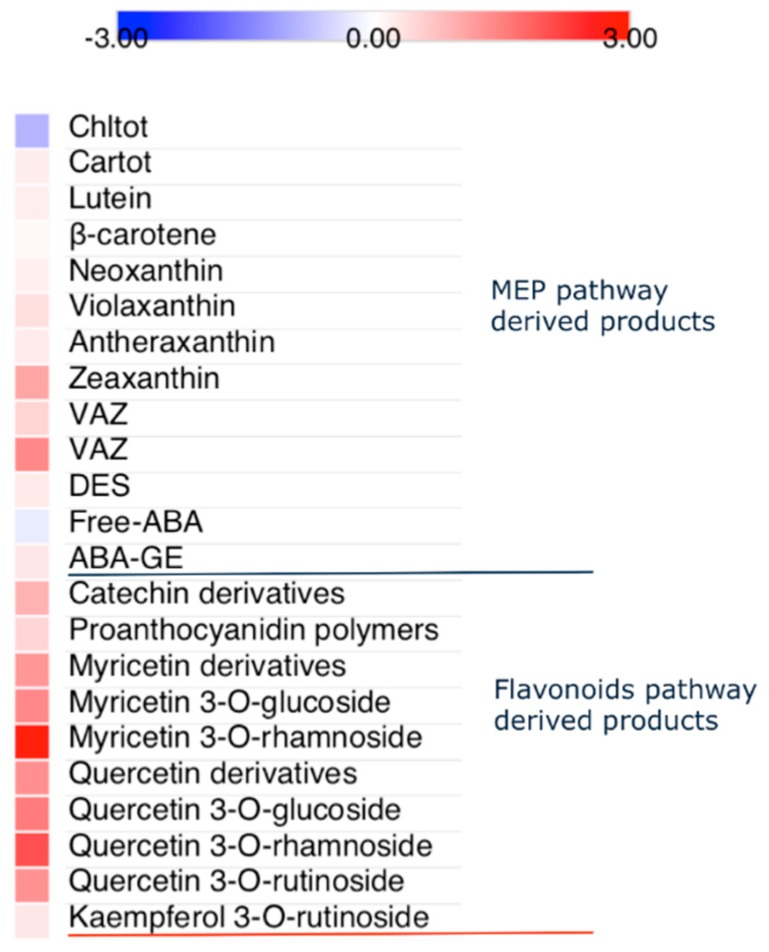
Changes in isoprenoid and flavonoid-derived products in *C. incanus* leaves adapted to shade or full sunlight. Fold changes, on log_2_ basis, denotes sun to shade metabolite concentration (on leaf dry mass basis).

**Table 1 ijms-20-03599-t001:** Relevant morpho-anatomical and physiological-related traits in *C. incanus* adapted to full shade or fully exposed to sunlight. Leaf water potentials were measured at predawn (PD) and midday (MD). Gas exchange measurements were conducted between 12:00–13:00 h, whereas chlorophyll fluorescence-derived parameters originate from measurements performed between 12:00–14:00 h, during July. Data are means ± standard deviation (*n* = 5 for ψ_w_, A_N_, and g_s_, *n* = 4 for F_v_/F_m_, Φ_PSII_, ETR, 1−*q*_P_, and NPQ). Asterisks denote significant differences at 5% level, using Tukey’s test.

Trait	Shade	Sun
Leaf thickness (μm)	145.3 ± 12.5 *	267.0 ± 18.5
Leaf mass per area (g·dw·m^−2^)	6.2 ± 0.8 *	10.8 ± 1.1
Water potential (ψ_w_^PD^,−MPa)	−2.12 ± 0.22 *	−2.95 ± 0.32
Water potential (ψ_w_^MD^, −MPa)	−2.85 ± 0.32 *	−3.63 ± 0.35
Net photosynthesis (*A*_N_, μmol CO_2_ m^−2^·s^−1^)	1.16 ± 0.33	1.28 ± 0.56
Stomatal conductance (g_s_, mmol H_2_O m^−2^·s^−1^)	33.3 ± 4.3 *	48.9 ± 6.6
Intercellular CO_2_ concentration (μmol·mol^−1^)	256 ± 19 *	324 ± 27
Maximal PSII photochemistry (F_v_/F_m_)	0.81 ± 0.01 *	0.72 ± 0.02
Actual PSII photochemistry (Φ_PSII_)	0.56 ± 0.05 *	0.14 ± 0.02
Electron transport rate (ETR, μmol·e^−^·m^−2^·s^−1^)	47.2 ± 9.5*	120.4 ± 10.7
ETR/*A*_N_ (μmol·e^−^·μmol^−1^·CO_2_)	40.7 ± 8.6 *	94.1 ± 11.1
Excess excitation energy (1 – *q*_P_)	0.32 ± 0.05 *	0.75 ± 0.12
Nonphotochemical quenching (NPQ)	1.79 ± 0.07 *	2.98 ± 0.04

**Table 2 ijms-20-03599-t002:** RNA sequencing and assembly statistics of *C. incanus* transcriptome.

Item	Number
Read length (bp)	100
Number of paired-end reads	36,988,026
Total trinity transcripts	59,030
Percent GC	43.01
Contig N50 (bp)	1462
Median contig length (bp)	728
Average contig (bp)	1028
Total assembled bases	60,727,583

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
