# Peer review of "Dissecting Adaptation Mechanisms to Contrasting Solar Irradiance in the Mediterranean Shrub *Cistus incanus"

_ijms, 2019, doi:10.3390/ijms20143599_

Round 1
Reviewer 1 Report
The manuscript by Sebastiani et al. is devoted to the interesting problem – analysis of mechanisms of adaptation of Mediterranean shrub Cistus incanus to light conditions and temperature. The work is potentially interesting; however, there are same question and remarks.
1. Table 1: Were there significant differences between leaf thickness, leaf mass per area or intercellular CO2 concentration? These differences seem to be significant (especially, for leaf thickness).
2. Lines 101-102: Can these temperature difference (34.9°C - 31.8°C = 3.1°C) influence physiological process in investigated plant? If yes then there were complex influence (temperature + excess light). It should be clarified.
3. Lines 440-442: What were conditions of CO2 assimilation measurements? Was artificial light used? What were CO2, H2O and temperature controlled? What were mean of the values? Was sunlight or artificial light used?
4. Lines 443-454: Details of PAM measurements should be clarified. How much saturation pulses were used? What were time intervals between the pulses? What was duration of the actinic light illumination before estimation of NPQ, ФPSII, ETR? Were the parameters stationary? Or some time point of dynamics was used?
5. Were measurements of gas-exchange (Li-Cor) and PAM-measurements at the same time?
6. Lines 453-454: Was coefficient “0.5” correct? The portion of light energy, distributed to PSII (dII) can be strongly varied (Plant Cell Physiol, 2005, 46, 629–637; Planta, 2012, 235, 819–828; Photosynth Res, 2013, 117, 529-546.); moreover, dII can be decreased from about 0.5 to about 0.4 with increase of duration of illumination (Front Plant Sci, 2015, 5: 766) and can be regulated by stressor actions (e.g. the dII was increased after local action of heating, Photosynth Res, 2018, 136, 215-228). Thus, it is not clear – can changes in dII participate in changes in ETR and assimilation in plants with different sunlight conditions? Decrease in dII can be protective mechanism under excess light. There are simple methods of estimation of dII (on basis of ФPSII, light intensity, CO2 assimilation and respiration, Plant Cell Physiol, 2005, 46, 629–637; on basis of ФPSI and ФPSII at weak light intensity, Front Plant Sci, 2015, 5: 766; on basis of ФPSI and ФPSII at high light intensity, Photosynth Res, 2013, 117, 529-546.) It is probable that estimation of dII can be important additional result to this work. Thus, I suppose that problem of dII, at least, should be discussed in the manuscript. It can improve analysis of photosynthetic changes in the manuscript.
7. Lines 457-459: I suppose that brief description of this protocol will be useful for readers.
Thus, I suppose that revision is necessary.
Author Response
First, thanks for the comments, irrespective of their nature. We think this revised version of the MS is better than the previous one thanks to your constructive criticism.
You are right about the statistics in table 1. We forgot simply to include statistical significance in some traits in Table 1. Now they have been included.
There is an interaction effect of water deficit and temperature on photosynthesis. We have explained in depth, see discussion red characters, the relationship between temperature and water stress, and photosynthesis in our experiment.
We have included details of both gas exchange and chlorophyll fluorescence analyses. I also agree with the referee comments about the distribution of light energy between PSII and PSI under severe excess of light, although my expertise is most on secondary metabolism. Unlikely we can assume an equal distribution. However, our analysis did not allow estimate the extent to which light distribution decreased in sun-adapted leaves in our experiment. Consequently, we have commented this fact, well explaining the limitations due to our measurements. We have also highlighted when gas exchange and PAM measurements were performed: always during the midday hours, from 12-13 for gas exchange and 12-14 for PAM (see also caption in Table 1).
Reviewer 2 Report
The manuscript by Sebastiani et al. reports the biochemical and molecular differences between limited and excessive solar radiation upon Cistus incanus. All the aspects investigated here have been reported before. Overall the manuscript provides limited novel information about the adaptation mechanisms underlying different light density.
I have several concerns (see the details below) and would like authors to address these issues.
1 For the realtime PCR results as shown in Figures 1 and 2, the statistic analysis should be applied to the difference between shade and sun leaves.
2 in line 174, the sucrose synthesis and regulation related genes were analyzed suddenly. In the introduction the roles of sucrose should be mentioned so that we know why they are analyzed in the results.
3 Tables are used to show the differences of gene expression from the RNA-seq data. I suggest using the heatmap to show these data so that they could be clearly seen. Same for the metabolites analysis (it is better to use column chart).
4 I think there should be a Figure to show the overall function analysis of the different expressed genes between two conditions. In this case, we can see in what pathway the genes are greatly different expressed. Then we have an idea why the authors chose the following pathways to discuss, such as light response, sucrose synthesis and transport, and photooxidative stress protection.
5 In this study, the RNA-seq data could be fully utilized. Some novel regulators could be identified through these data. Some molecular experiments could be conducted to study the function of the novel regulators.
Author Response
The revision of our MS according to the reviewer comments, with the inclusion of heat maps for both DEG and metabolites, has required a great effort. In fact, both the Results and the Discussion sections have been largely changed. We have included a section regarding transcription factors and their potential functional roles in the acclimation of plants to light excess. Revisions have been typed in blue characters. We also have included additional references, and clearly indicated the statistical significance in qt-rt PCR. Our discussion has been also changed trying to highlight major pathways involved in our plants under contrasting light irradiance. This can be observed in our new Conclusions section, which has been largely re-written. We have commented about the up-regulation of genes regulating shoot branching in sun leaves. While we are sure about the serious revision work we did, we hope it has improved the scientific quality of the MS to merit publication in IJMS.
Finally, we have found very difficult to include in the Introduction the issue concerning carbohydrates, which was indeed in some way an unexpected result of our study. The inclusion of this issue should start considering different water potential and osmotic adjustment in sun and shade leaves, and the role of carbohydrates in osmotic adjustment. Then we could take into account the negative role of soluble carbohydrates accumulation on photosynthesis. In other words, the flow, if any, in the Introduction might be lost. Nonetheless, we hope this has been adequately addressed in the Discussion.
Round 2
Reviewer 2 Report
Massimiliano Tattini and collaborators have done a very nice job in response to my comments and requests. Also, I found that the authors have done great revision based on other reviewers' comments. I really appreciate the time and effort that the authors have put in over the past two months. I am satisfied with their reply and the new results in this revised manuscript.